



# Global Trends of Large Wind Farm Performance based on High Fidelity Simulations

Søren Juhl Andersen[1], Simon-Philippe Breton[2,3], Björn Witha[4,5], Stefan Ivanell[2], and Jens Nørkær Sørensen[1]

[1]DTU Wind Energy, Technical University of Denmark, 2800 Lyngby, Denmark.
[2]Uppsala University, Dep. of Earth Sciences, Campus Gotland, Cramérgatan 3, 62 157 Visby, Sweden.
[3]Environment and Climate Change Canada, 2121 Route Transcanadienne, Dorval, Québec, H9P 1J3, Canada.
[4]ForWind - Center for Wind Energy Research, Carl von Ossietzky University Oldenburg,Küpkersweg 70, 26129 Oldenburg,Germany.
[5]energy meteo systems GmbH, Oskar-Homt-Str. 1, 26131 Oldenburg, Germany.

**Correspondence:** Søren Juhl Andersen, sjan@dtu.dk

**Abstract.** A total of 18 high fidelity simulations of large wind farms have been performed by three different institutions using various inflow conditions and simulation setups. The setups differ in how the atmospheric turbulence, wind shear and wind turbine rotors are modelled, encompassing a wide range of commonly used modelling methods within the LES framework. Various turbine spacings, atmospheric turbulence intensity levels and incoming wind velocities are considered. The work

performed is part of the International Energy Agency(IEA) wind task Wakebench, and is a continuation of previously published results on the subject. This work aims at providing a methodology for studying the general flow behavior in large wind farms in a systematic way. It seeks to investigate and further understand the global trends of wind farm performance, with a focus on variability.

Parametric studies first map the effect of various parameters on large aligned wind farms, including wind turbine spacing,

wind shear and atmospheric turbulence intensity. The results are then aggregated and compared to engineering models as well as LES results from other investigations to provide an overall picture of how much power can be extracted from large wind farms operating below rated level. The simple engineering models, although they cannot capture the variability features, capture the general trends well. Response surfaces are constructed based on the large amount of aggregated LES data corresponding to a wide range of large wind farm layouts. The response surfaces form a basis for mapping the inherently varying power

characteristics inside very large wind farm, including how much the turbines are able to exploit the turbulent fluctuations within the wind farms and estimating the associated uncertainty, which is valuable information useful for risk mitigation.

## 1 Introduction

As renewable energies are expected to take an increasing share of the future electricity production, wind energy is progressing, where wind farms are being built in ever increasing sizes, especially offshore. Wind turbines operating far downstream in very

large farms are subject to complex flow conditions, comprised of the combined interaction between the atmosphere and the complex wake dynamics introduced by the wind turbines. Several factors come into play and contribute to the complexity of





the farm flow. These factors can be grouped into atmospheric conditions (*e.g.* stability, shear, veer and turbulence intensity), wind farm conditions (turbine size, farm layout) and combined effects as the turbines affect the atmospheric flow. A better understanding of how the flow develops in large wind farms is crucial in order to better plan and control the wind farms, and

to optimize their production.

Almost ten years ago, Barthelmie et al. (2009) observed that several CFD wake models performed adequately to predict wake losses in small wind farms, whereas they seemed to underpredict wake losses in very large wind farms. The latter was also observed shortly after by Rathmann et al. (2010). This was explained in both cases by a lack of these models to account for the effect that large wind farms are expected to have on the atmospheric boundary layer, which can lead for example to

a modification of the vertical wind profile. The development of the flow is indeed very different for small wind farms when compared to large ones. As pointed out by Calaf et al. (2010), the difference between the upstream and downstream kinetic energy fluxes determines the power extracted by a single turbine, while for a turbine operating in a so-called fully developed wind-turbine array boundary-layer, the kinetic energy has to be entrained from the flow above. Under such conditions, in a regime that can be defined as asymptotic, the important exchanges occur in the vertical direction. The fully developed flow

regime is obtained for wind farms whose length exceeds the height of the atmospheric boundary layer by an order of magnitude according to Calaf et al. (2010). Johnstone and Coleman (2012) define this coupling with the atmospheric boundary layer as a two-way process, arguing that an understanding of the behaviour of the arrays depends on a complementary understanding of the associated atmospheric boundary layer.

This complex wake problem has attracted the interest of numerous researchers for many years, where work has been per-

formed using several numerical methods, including both engineering type models, such as those by Jensen (1983) and Frandsen and Madsen (2003), and high fidelity Large Eddy Simulations (LES), as well as measurements, both model and full scale.

Stevens et al. used LES to study the effect of turbine spacing on the power output of large wind farms Stevens et al. (2015b). They showed that the power output in the fully developed regime for a staggered wind farm depends mostly on the geometric mean of the streamwise and spanwise turbine spacings, while it depends mostly on the streamwise spacing for an aligned

wind farm. They also mentioned that the assumptions associated with effective roughness height models are more adapted to staggered wind farms than aligned ones. The power output in the farms was further found by Stevens et al. (2015b) to be well correlated with the vertical kinetic flux, in accordance to what is obtained by Calaf et al. (2010), who used LES of large wind farms to quantify the vertical transport of momentum and kinetic energy across the boundary layer. Stevens et al. (2015a), in a different work, also developed a so-called coupled wake boundary layer model of wind farms to predict the power output

in large wind farms. The model coupled a wake model for the turbines with a top-down boundary layer model. The coupled model is much simpler and faster than LES, and was shown to compare well with averaged LES results. LES was further used by Stevens (2016) to study how the optimal wind turbine spacing depends on the wind farm length to find that it is remarkably larger for large wind farms when compared to smaller, conventional wind farms.

Yang et al. (2012) used LES to investigate the effect of streamwise and spanwise turbine spacing on the power output

and turbulence intensity in infinite aligned wind farms. They reached the same conclusion as Stevens et al. (2015b), *i.e.* that using a larger streamwise spacing is more efficient than using a larger spanwise one in increasing the power extraction of an





aligned wind farm. Based on their study, they suggested an improved effective roughness height model taking into account the various effects of the spacings in these two directions. Yang and Sotiropoulos (2014) studied infinite staggered wind turbine arrays using the same method. The wake behaviour, which was found to be significantly different from the aligned cases, was classified into three wake patterns depending on how a given turbine wake interacts with the turbine wakes downstream.

Wu and Porte-Agel (2013) used LES to study turbulent flow inside and above large wind farms considering a neutral boundary layer and both an aligned and a staggered layout. Wind turbine rotors were modelled using various forms of actuator discs. An important influence from the wind farm configuration was found in terms of the turbulent flow inside and above the wind farm, and the staggered configuration was shown to be more efficient to extract momentum from the flow.

Breton et al. (2014) performed LES to study the influence of imposing turbulence on the asymptotic wake deficit along a row of 10 turbines modelled as rotating actuator discs. An asymptotic wake state appeared to be reached near the end of the 10 turbine row when looking at for example the average of the standard deviation of the velocity components, turbulence kinetic energy and mean power that then became more or less independent of the downstream position. Higher turbulence intensity levels made changes towards this state happen faster. Andersen et al. (2016) found that the asymptotic state is reached by the 5th or 6th turbine.

Andersen et al. (2015) worked towards quantifying the variability in LES of very large wind farms modelled as actuator discs or actuator lines, pointing out that LES are inherently dynamic, and that performing simple averages of the various physical quantities does not capture the dynamics, which can lead to misleading interpretations when comparing various LES models with each other or with experimental results.

Cal et al. (2010) used Particle-image-velocimetry (PIV) on an array of scaled model wind turbines to show that the power extracted by the wind turbines is of the same order of magnitude as the fluxes of kinetic energy that are related to the Reynolds shear stresses. This serves as an experimental proof of the importance of vertical transport in the boundary layer, as is also obtained in various LES works mentioned above. Newman et al. (2014) also employed PIV on a scaled model wind farm and found that the majority of the entrainment originates from scales larger than the turbine size. The analysis was extended in the numerical work by Andersen et al. (2017b), who showed that the large dominant length scales are associated with and limited by the turbine spacing.

The aim of the present article is to present a methodology that can be used in a systematic way to further understand the general flow behaviour in large wind farms. As outlined above, a number of research groups are today frequently simulating the flow in large wind farms using high fidelity methods to further understand basic flow features. However, since there is a large variability of parameters, e.g., flow directions, choice of verification cases with different turbine spacings, atmospheric conditions etc., it is often very difficult to draw general conclusions through direct comparisons. The aim with the developed methodology is to capture key parameters from different setups to be able to investigate the global trends of wind farm performance. Here, results from high fidelity simulations are combined and systematically analyzed. As will be shown in this article, the quality of the conclusions that can be drawn depends on the extent of data that can be used. By quantifying the variability for different situations the uncertainty can be estimated.





In the present work, data derived from LES will be used, as this kind of high fidelity data has been shown to produce very reliable results as regards to the development of the flow within wind farms, see *e.g.* Breton et al. (2017). In the present work, three different research groups are contributing with input. This results in an improved understanding of the big picture and how production depends on turbine separation, flow angles and atmospheric conditions.

The work is a continuation of previous work that studied the variability of the flow statistics in LES performed on large wind farms by Andersen et al. (2015). A more general analysis is performed here, where a greater quantity of results obtained under different configurations are considered. The focus is still on variability, with an emphasis on wind power, where the effect from various parameters like turbulence intensity and wind turbine spacing is studied. While only aligned wind farms have been simulated for this study, results obtained from staggered cases already published by other researchers are included for completion. Furthermore, the large number of turbine spacings and farm configurations considered in this work is believed to cover the conditions associated with both staggered and aligned cases as the simulated wind farms do not only have rectangular layouts.

The paper is arranged as follows: in section 2, the methodology used to perform this work in terms of numerical methods is outlined, followed by the simulation setups considered to run each of these methods in section 3. Results are then presented and discussed in section 4, where works from other researchers are also included, before the main conclusions from the work are summarized and discussed in section 5.

## 2 Methodology

In this section, an overview of the main differences as regards the methodology used by the different participants is provided. Detailed information on the theoretical background associated to each method can be found in the publications that are referred to.

### 2.1 Numerical Solvers

Results from two different CFD codes are used.

#### 2.1.1 EllipSys3D

EllipSys3D is a 3D flow solver that was developed at DTU, Michelsen (1992), and the former Risø, Sørensen (1995). It solves the discretized incompressible Navier-Stokes equations in general curvilinear coordinates using a block structured finite volume approach. It is formulated in primitive variables (pressure-velocity) in a collocated grid arrangement. Additional details about this code can be found in Mikkelsen (2003) and Troldborg (2008).

#### 2.1.2 PALM

PALM (Parallelized LES Model) was developed at Leibniz Unversity Hannover and has been applied several years for the simulation of a variety of atmospheric and oceanic boundary layers. Recently, it has been enhanced by a wind turbine model,





see Witha et al. (2014). It is an open source, highly parallelized LES model which solves the filtered, incompressible, non-hydrostatic Navier-Stokes equations under the Boussinesq approximation on an equidistant Cartesian grid. The sub-grid scale turbulence is parameterized by a 1.5th order closure after Deardorff (1980). Further details about this code can be found in Maronga et al. (2015).

## 2.2 Turbine modelling

### 2.2.1 EllipSys3D

The wind turbines are modelled by DTU and Uppsala University (UU) by using the actuator line (AL) and actuator disc (AD), respectively. In the former, body forces are distributed along rotating lines, while they are distributed along a rotating disc in the latter. Details about the implementation of the AD and AL in EllipSys3D can be found in Mikkelsen (2003) and Sørensen
and Shen (2002), respectively. Local blade forces are determined using tabulated airfoil data and the local inflow conditions. In the DTU-AL model, the 2D airfoil data are corrected for 3D effects, see *e.g.* Hansen et al. (2006). The body forces in the DTU implementation of the AL are further calculated through a coupling with Flex5, which is a full aeroelastic code used for calculating deflections and loads on wind turbines. Øye (1996) provides details on Flex5 while Sørensen et al. (2015) describes the coupling.

### 2.2.2 PALM

The PALM implementation considers an AD model with rotation (FW-AD-R) in which local body forces are derived from airfoil data. The PALM simulations were performed by ForWind (FW). In contrast to the AL method, the forces are distributed across the rotor plane. This model also includes tower and nacelle effects that are modelled by a drag force approach. See Dörenkämper et al. (2015) for details of the PALM implementation.

### 2.2.3 Turbine Controller

The three models used in this work include a turbine controller. This causes the applied body forces to be governed by the inflow conditions, meaning that the turbines are not constantly loaded, but operate as "real turbines". Larsen and Hanson (2007) or Hansen et al. (2005) provide a general description of such controllers.

### 2.2.4 Turbine Data

Two different three-bladed horizontal axis wind turbines have been considered in the simulations, *i.e.* the NM80 and the NREL 5MW. The NM80 turbine, see *e.g.* Aagaard Madsen et al. (2010), has a radius $R$ of 40 m, a hub height $z_{hub}$ of 80m, and a rated power of 2.75 MW at a nominal hub height velocity of $14 \, \mathrm{ms}^{-1}$. The radius of the NREL 5MW turbine is 63 m, its hub height is 90 m, and its rated power is 5 MW at $11.4 \, \mathrm{ms}^{-1}$, see Jonkman et al. (2009). Figure 1 compares the $C_P$ and $C_T$ of the two turbines, which are comparable although the $C_T$ is higher for the NREL 5MW than for the NM80 for below rated.


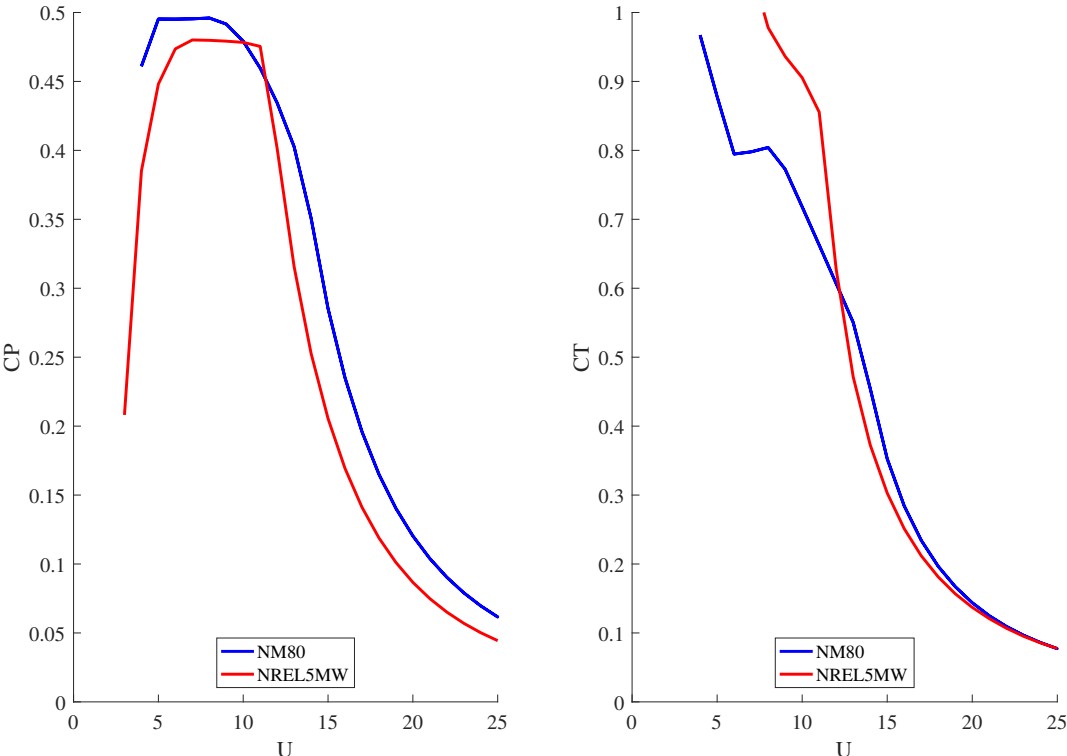

**Figure 1.** $C_P$ and $C_T$ curves for the NM80 and NREL 5MW turbines.

## 3 Simulations Setup

In the coordinate system used in this work, $x$, $y$ and $z$ correspond respectively to the streamwise, crosswise and vertical directions. The grids used for the simulations are equidistant in the horizontal direction in all cases. The grids are usually stretched in the vertical direction from a significant distance above the wind turbines.

### 3.1 Atmospheric Boundary Layer and Turbulence

All participants simulated a neutrally stratified atmospheric boundary layer (ABL). Details about the methods used to model the ABL and associated turbulence in respectively EllipSys3D and PALM are provided below.

#### 3.1.1 EllipSys3D

EllipSys3D uses the prescribed boundary layer (PBL) method, in which body forces are used to impose any arbitrary vertical wind shear profile, see Mikkelsen et al. (2007) and Troldborg et al. (2014). A comparison of PBL with a wall model approach was performed by Sarlak et al. (2015). This study showed that these two approaches yield very comparable vertical profiles of mean streamwise velocity, shear stress, and streamwise velocity fluctuations in the rotor region when large wind farms are





modelled. Ambient turbulence is modelled by introducing pregenerated synthetic ambient turbulence using the Mann model, see Mann (1998). Turbulence planes are imposed at an axial position of 6 $R$ and 13 $R$ in the DTU and UU simulations, while
the first simulated turbine is located at 10 $R$ and 30 $R$ from the inlet, respectively.

### 3.1.2 PALM

PALM uses a no-slip bottom boundary condition and the Monin-Obukhov similarity theory between the surface and the first grid level to model the atmospheric boundary layer. Random perturbations are initially imposed on the velocity fields until atmospheric turbulence has developed in a precursor simulation. The latter is performed on a smaller domain with periodic
boundary conditions in the streamwise and lateral directions. The precursor results are used to initialize the full simulations with non-periodic boundary conditions in the streamwise direction. Turbulence recycling is also applied, see Maronga et al. (2015) for details.

### 3.1.3 Summary of Numerical Methods

An overview of the numerical methods described in the previous sections are summarized in Table 1 for each of the three
contributions.

**Table 1.** Summary of methods.

| Method | $DTU$ | $FW$ | UU |
|---|---|---|---|
| CFD Solver | EllipSys3D | PALM | EllipSys3D |
| Coriolis | No | Yes | No |
| Turbine | NM80 | NREL 5MW | NREL 5MW |
| Turbine Modelling | Actuator Line | Actuator Disc | Actuator Disc |
| $R$ | 40 m | 63 m | 63 m |
| $z_{hub}$ | 80 m | 90 m | 90 m |
| Controller | Yes | Yes | Yes |
| 3D effects correction | Yes | No | No |
| Aero-elastics included | Yes | No | No |

### 3.2 Overview of simulations considered

A total of 18 large wind farms have been simulated and analyzed. The majority of the simulations are performed for below rated conditions at approximately 8 m/s for a range of ambient turbulence intensities($0-15\%$) and turbine spacings($12R-20R$) in streamwise and lateral direction. Additionally, two simulations with 15 m/s are included, which corresponds to just above rated.
The simulations are summarized in Tables 2, 3, and 4 for the contributions from DTU, UU, and FW, respectively. Noticeably,



the simulation differences are particularly related to the difference in modelling the atmospheric boundary layer, which gives different shear velocity profiles. The DTU simulations have previously been analyzed in terms of flow statistics and distribution in Andersen et al. (2016).

**Table 2.** Overview of simulations performed by DTU. The simulations include 16 turbines and 60 mins of data.

| Name | $U_0$ | Ambient TI | Shear | Turbine Resolution $[R]$ | Spacing ($S_X \times S_Y$) |
|------|-------|-----------|-------|--------------------------|----------------------------|
| $DTU1$ | 8 m/s | 0% | 0.14 | 0.0625 | $12R \times 20R$ |
| $DTU2$ | 8 m/s | 3% | 0.14 | 0.0625 | $12R \times 20R$ |
| $DTU3$ | 8 m/s | 15% | 0.14 | 0.0625 | $12R \times 20R$ |
| $DTU4$ | 15 m/s | 15% | 0.14 | 0.0625 | $12R \times 20R$ |
| $DTU5$ | 8 m/s | 0% | 0.14 | 0.0625 | $12R \times 12R$ |
| $DTU6$ | 15 m/s | 0% | 0.14 | 0.0625 | $12R \times 12R$ |
| $DTU7$ | 8 m/s | 0% | 0.14 | 0.0588 | $14R \times 14R$ |
| $DTU8$ | 8 m/s | 0% | 0.14 | 0.0625 | $20R \times 20R$ |

**Table 3.** Overview of simulations performed by UU. The simulations include 16 turbines and 30 mins of data. The vertical shear profile imposed within the PBL method is determined using the same equivalent roughness as the one used in the Mann algorithm to generate turbulence Mann (1998)

.

| Name | $U_0$ | Ambient TI | Equivalent roughness | Turbine Resolution $[R]$ | Spacing ($S_X \times S_Y$) |
|------|-------|-----------|----------------------|--------------------------|----------------------------|
| $UU1$ | 8 m/s | 15% | $0.5m$ | 0.0781 | $8R \times 20R$ |
| $UU2$ | 8 m/s | 15% | $0.5m$ | 0.0781 | $12R \times 20R$ |
| $UU3$ | 8 m/s | 15% | $0.5m$ | 0.0781 | $14R \times 20R$ |
| $UU4$ | 8 m/s | 15% | $0.5m$ | 0.0781 | $20R \times 20R$ |

## 4   Results and Discussion

The present analysis is an extension of the previous work on the inherent variability of the flow statistics in LES as presented by Andersen et al. (2015). The long term average velocity within large wind farms is expected to converge towards a constant level deep inside the wind farm, where a balance between the extracted energy and the entrained energy is reached. However, as shown by Andersen et al. (2015) the distributions of instantaneous and even 10 min average velocities show significant variability within the same simulation. Here, the focus of the present study is on mechanical power, as opposed to the electrical

power, which requires estimation of the electrical losses in for instance the generator. Hence, the power production calculated





**Table 4.** Overview of simulations performed by FW. The simulations included two rows of 50 turbines and 60 mins of data. Data is only given for one row of 50 turbines.

| Name | $U_0$ | Ambient TI | Equivalent roughness | Turbine Resolution $[R]$ | Spacing $(S_X \times S_Y)$ |
|------|-------|-----------|---------------------|--------------------------|----------------------------|
| $FW1$ | 8 m/s | 3% | $10^{-8}m$ | 0.127 | $6R \times 20R$ |
| $FW2$ | 8 m/s | 3% | $10^{-8}m$ | 0.127 | $12R \times 20R$ |
| $FW3$ | 8 m/s | 3% | $10^{-8}m$ | 0.127 | $20R \times 20R$ |
| $FW4$ | 8 m/s | 10% | $0.15m$ | 0.127 | $6R \times 20R$ |
| $FW5$ | 8 m/s | 10% | $0.15m$ | 0.127 | $12R \times 20R$ |
| $FW6$ | 8 m/s | 10% | $0.15m$ | 0.127 | $20R \times 20R$ |

as:

$$P_{mech} = T \cdot \omega$$

where $T$ is the torque and $\omega$ is the angular velocity.

The different numerical setups enable different parametric studies, where the effect of free stream turbulence intensity, of
turbulence and shear combined, as well as of turbine spacing is investigated.

Finally, the large amount of data is aggregated, and a more generalized analysis is performed on mechanical power production and variability within large wind farms.

### 4.1 Variability of LES

Simulations $DTU3$, $UU2$, and $FW5$ (*cf.* Tables 2, 3, and 4) are comparable in terms of freestream velocity at hub height,
turbulence intensity, and spacing. Box plots based on the 10 min average mechanical power production normalized by rated power $P_0$ of the first 16 turbines for are given in Figure 2. Box plots are a compact way to visualize the distribution in terms of meadian and the upper and lower quartiles. The 10 min averages have been calculated for the entire time series by shifting the averaging window by 1 min to increase the number of samples, *i.e.* a total of 51 samples from 60 min simulation time and 21 samples from 30 min simulation time. This approach yields more samples and hence a first indication of the distribution, albeit
not statistically independent.

The results from DTU and UU are very comparable in terms of level of mechanical power production, while the FW results are approximately 40% lower. This is consistent with the flow results presented in Andersen et al. (2015) and presumably mainly due to lower turbulence in the FW results as well as difference in shear and Coriolis. The figure endorses the previous findings of large variability within LES of large wind farms, although the filtering effects lower the variability in power
compared to velocity. Here, the mechanical power production can vary by $\pm 10\%$ or more around the median. However, there are distinct regions within the farm where the variability is higher. This is particularly evident for turbines 8-11 in the UU results. Another interesting spatial effect is seen in both the results from DTU and FW, where the median peaks at the 7th



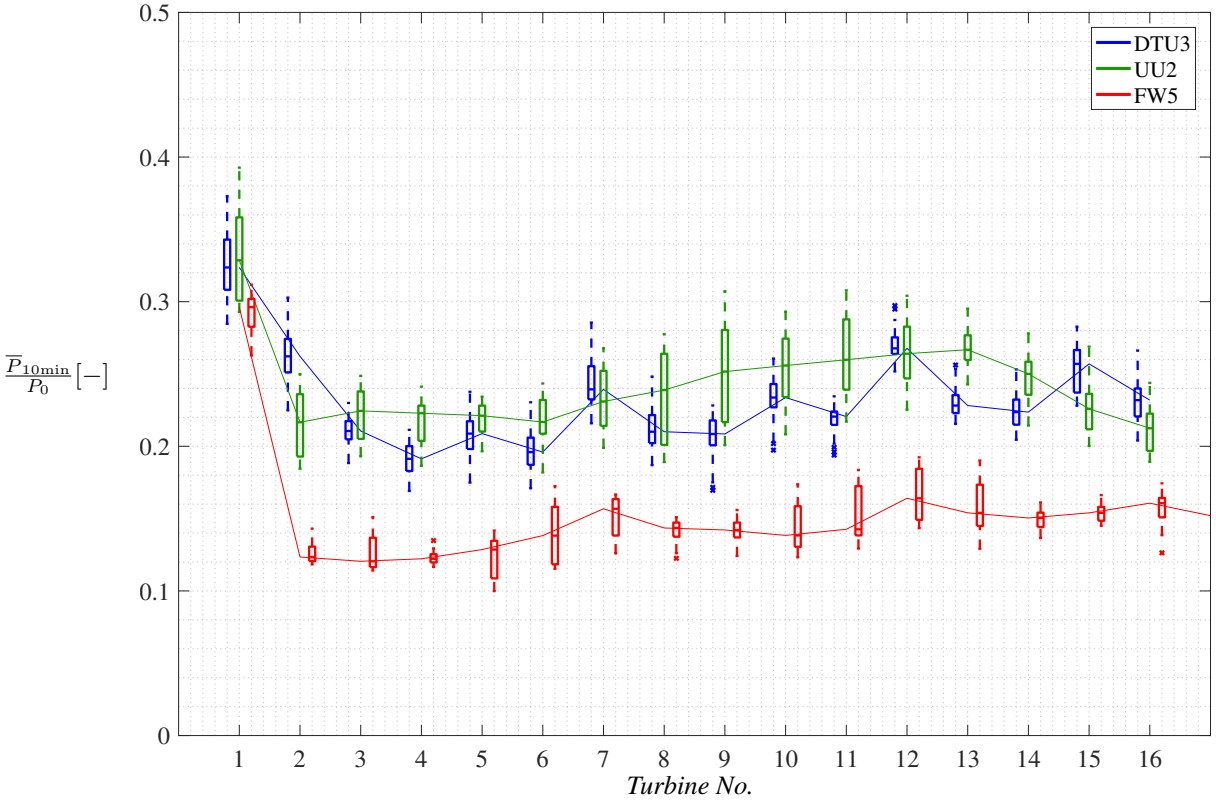

**Figure 2.** Box plots of the 10 min averaged mechanical power production normalized by rated power $P_0$ of the first 16 turbines in $DTU3$, $UU2$, and $FW5$. All simulations have $U_0 = 8$ m/s and $S_X = 12R$, and $S_Y = 20R$. The turbulence intensity is $TI \approx 15\%$ for $DTU3$ and $UU2$, while it is $TI \approx 10\%$ for $FW5$. Lines connect median values of all turbines. Outliers not shown for clarity.

and 12th turbine. This "anomaly" was first reported by Andersen et al. (2017a) based on analysis of the same simulations, where it was shown not to be related to the atmospheric turbulence. Given the difference in numerical setup, this corroborates

that the "anomaly" is a physical feature related to large scale physics dependent on turbine spacing, which needs additional investigation.

The simulations performed by FW included 50 turbines, so the full spatial extent of the wind farm is given in Figure 3. The "anomaly" appears throughout the wind farm with distinct peaks at turbines 7, 12, 16, 23, 30, 39, 42, and 45. Furthermore, the variability clearly increases towards the end of the wind farm, where the power production ranges from 0.13-0.20 of rated

power for the NREL 5MW turbine.

### 4.1.1    Effect of Turbulence Intensity

The simulations from DTU and UU utilize body forces to introduce ambient turbulence into the flow. This enables direct investigation of the isolated effects of changing the ambient turbulence by changing the forcing.

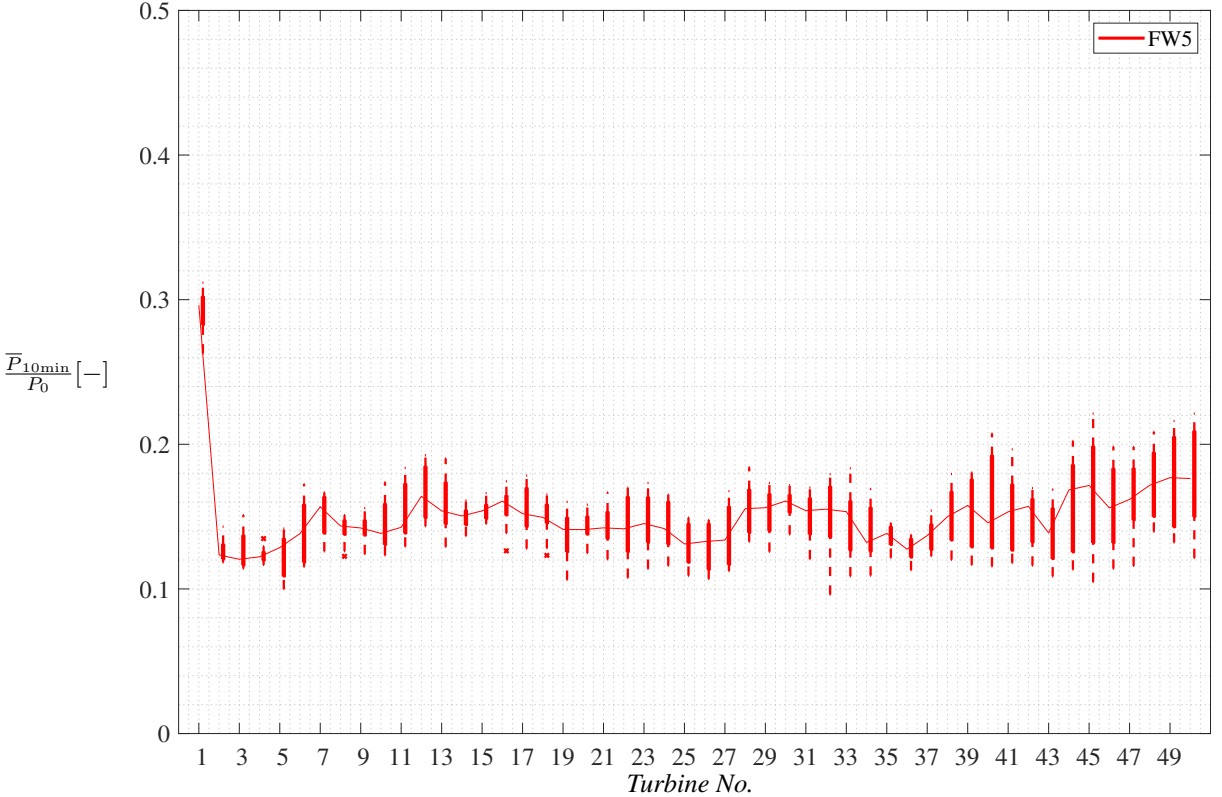

**Figure 3.** Box plots of $FW5$ for all 50 turbines. Outliers not shown for clarity.

Here, the distributions of instantaneous power production of the 16 turbines are compared directly in violin plots in Figure

4 for $DTU2$ and $DTU3$, *i.e.* identical setup except an approximate free stream turbulence of 3% and 15%, respectively. The

differences in the distributions are clear. An increase in freestream turbulence increases the mean level of power production due

to increased energy entrainment. Initially, the distributions are also broader for the high turbulent case than for the low turbulent

case, which appears Gaussian, in particular for the second turbine. The width of the distributions becomes more similar further

into the farm, but the difference in median level is maintained. Similar trends were reported by Andersen et al. (2016). The

effect of the controller is also clearly seen as the distributions are capped around $\frac{P}{P_0} \approx 0.33$.

#### 4.1.2 Effect of Shear and Turbulence Intensity

The simulations performed by FW include the combined effect of shear and turbulence intensity, as a change in equivalent

roughness yields different shear and turbulence profiles. Figure 5 shows violin plots of the instantaneous mechanical power

production in $FW2(TI = 3\%)$ compared to $FW5(TI = 10\%)$ for the first 16 turbines normalized by rated power. The distri-

bution is once again significantly broader for the high turbulence case and the distribution for the second turbine in the lower

turbulence case is close to Gaussian. However, the median level appears to be very similar for the following turbines(3-6) with





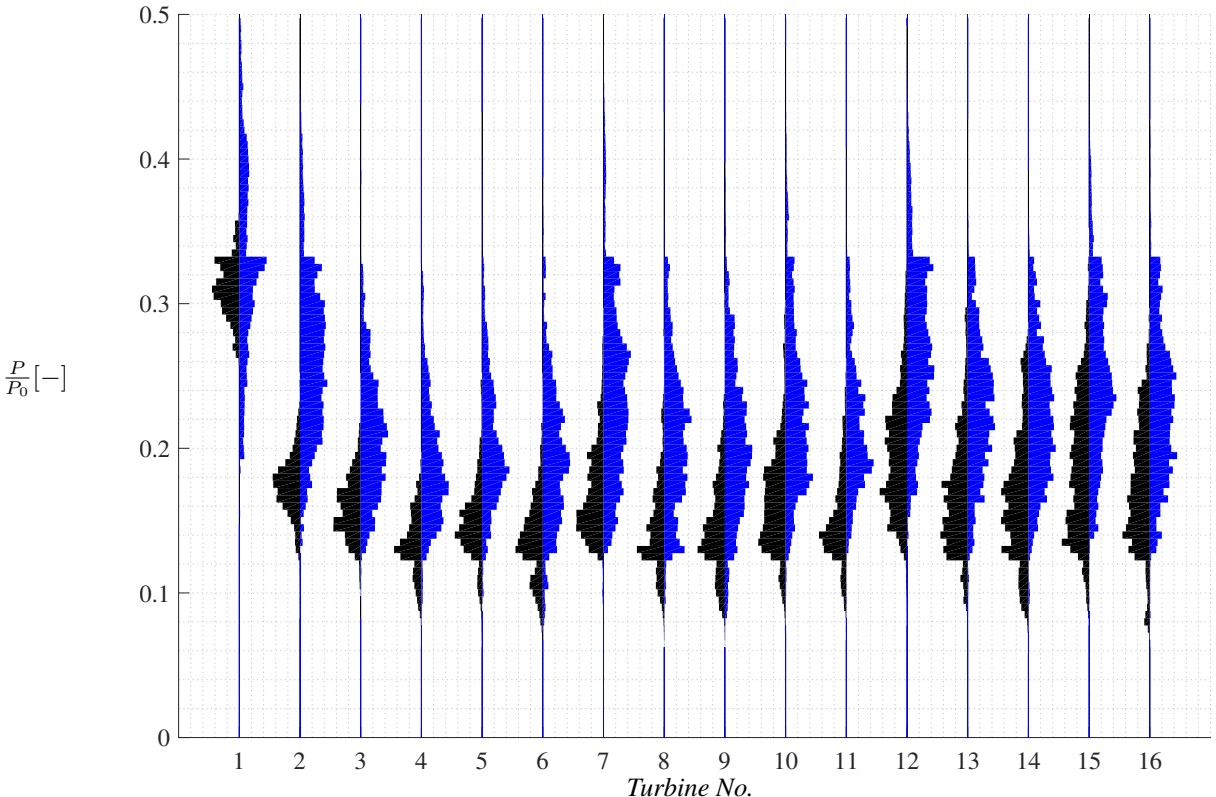

**Figure 4.** Violin plots comparing the influence on turbulence intensity on the instantaneous power production in $DTU2$ (black) and $DTU3$ (blue).

infrequent higher tails. Further into the farm, the distributions become broader for the high turbulent case with a slight increase in the median level. However, the increase in the median level is not as pronounced as in Figure 4, which indicates that high shear decreases the effects of an otherwise high turbulence intensity.

### 4.1.3 Effect of Spacing

Fig. 6 shows violin plots of the instantaneous mechanical power production in DTU5 compared to DTU7 for the first 16 turbines normalized by rated power. This allows comparing spacings of respectively $12R \times 12R$ and $14R \times 14R$. The fact that these simulations consider a zero level of incoming turbulence intensity explains the small spread of power values around the mean for the first turbines in the farm. The distributions broaden as the turbulence produced by the turbines themselves dominates further into the farm. As expected, a larger spacing is associated with greater values of mean power, as it allows more time for the wake flow to mix with the outside flow in between the turbines and to recover. The power distribution associated with the greater spacing appears Gaussian for the most part, while the one related to the shorter spacing of $12R \times 12R$ is more irregular and seems to consist in two distinct parts, presumably due to how the turbine controller reacts to being in the near wake.



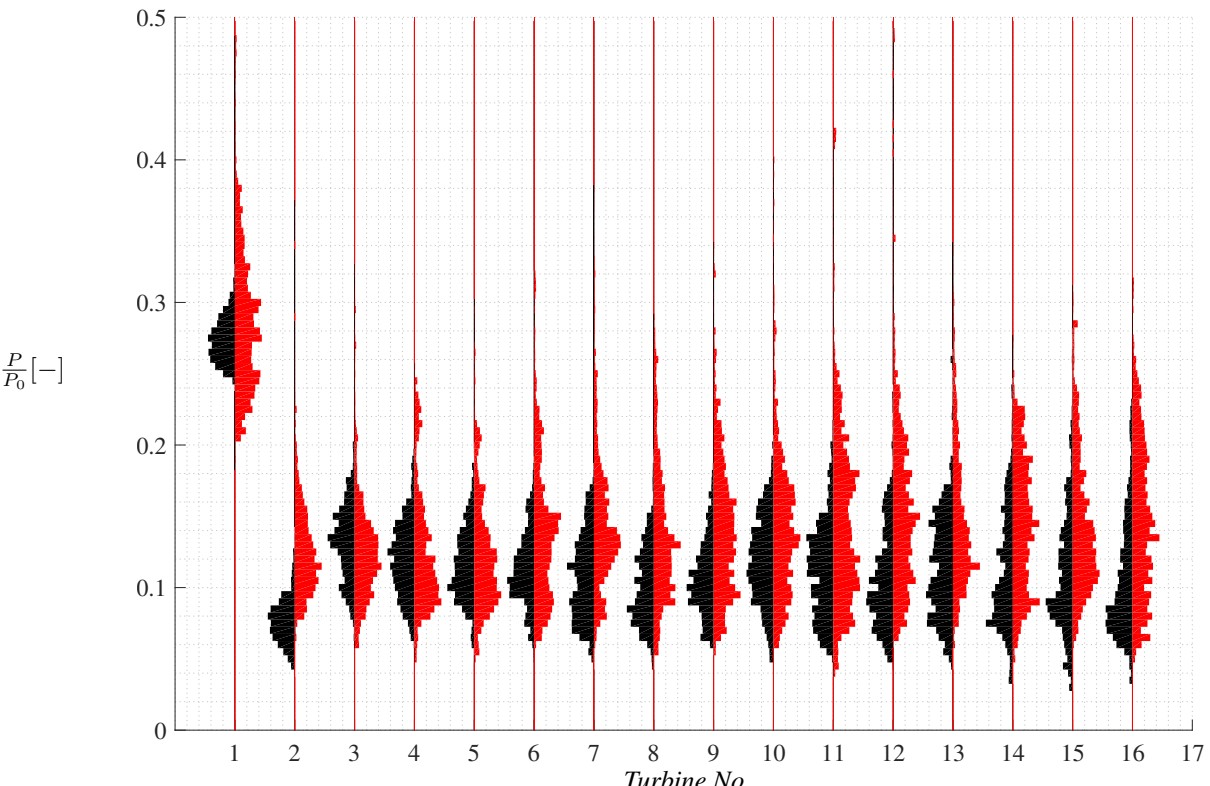

**Figure 5.** Violin plots comparing the influence of turbulence intensity and shear on the instantaneous power production in $FW2$ (black) and $FW5$ (red).

## 4.2 Aggregated Data

### 4.2.1 Comparison to Simple Engineering Models

The simulation data are aggregated in terms of 10 min statistics for each operating turbine. Aggregating the statistics from different simulations and numerical setups essentially assumes that all simulations are physically correct and correspond to different farms/turbines operating under different atmospheric conditions. The distributions have generally converged after the 6th turbine, despite the large variability, so the mean 10 min power production of all turbines from the 6th to the end of the row of all the 18 simulations are aggregated as representative of operating in "deep wind farm" conditions. The aggregated data is plotted in Figure 7 as function of a representative turbine spacing, $\sqrt{S_X \times S_Y}$, as suggested by Stevens et al. (2015a). The mean power production is normalized by the long term mean power production of the first turbine to enable a direct comparison with results taken from Stevens et al. (2015a). The data is colored according to inflow turbulence intensity and the symbols indicate if the results are from DTU, UU, or FW. The standard deviation for the different 10 min periods and turbines of the current simulations have been included as error bars.



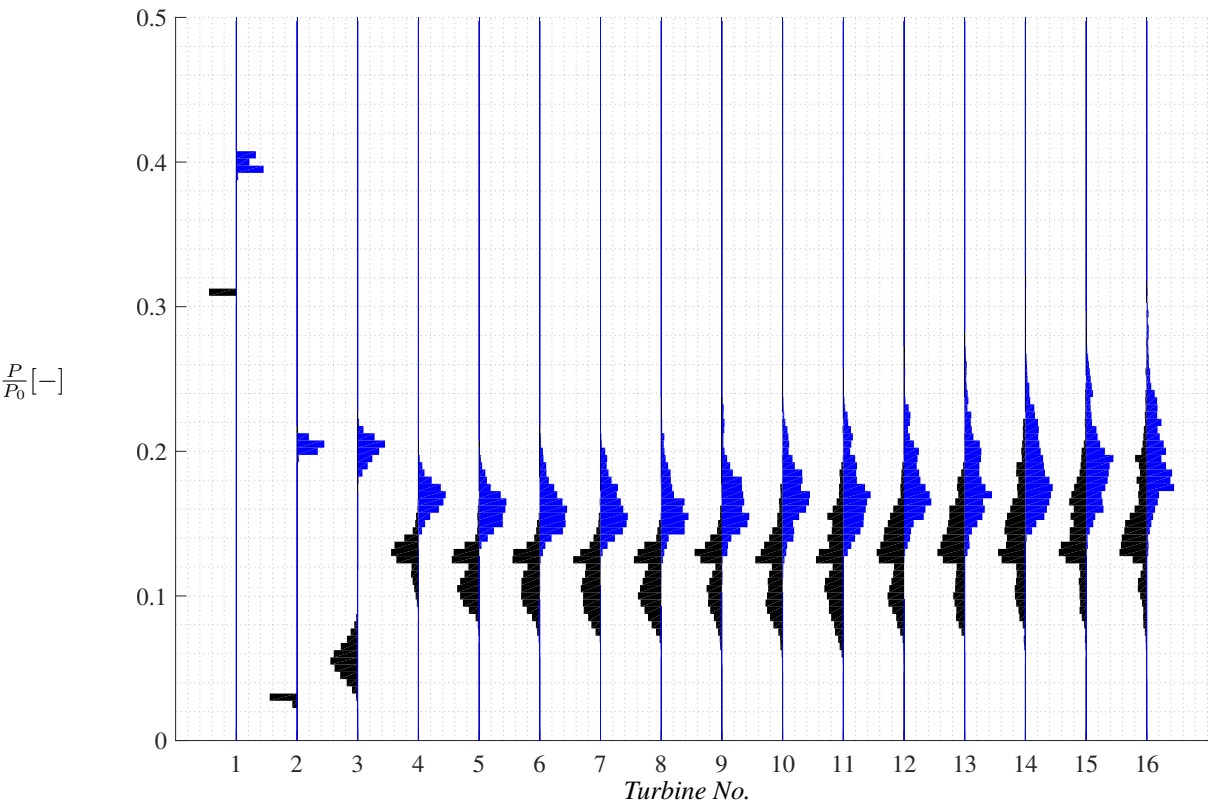

**Figure 6.** Violin plots comparing the influence of turbine spacing on the instantaneous power production in $DTU5$ (black) and $DTU7$ (blue).

It is clear how the 18 simulations follow the same trends as the data derived from Stevens et al. (2015a), that were obtained for both aligned and staggered configurations. Stevens et al. only present long term averaged values without the variability. The results are generally encompassed by the results of DTU and FW. All results fall within a clear limit showing how much power can be extracted from a wind farm operating below rated wind speed depending on representative turbine spacing.
The upper limit is indicated by $DTU4$ and $DTU6$, which have a freestream velocity above rated(15 m/s), but with different turbulence intensities. The power productions deep inside the farm result in below rated conditions for $DTU6$ due to no freestream turbulence, while the turbines in $DTU4$ also experience above rated velocities deep inside the farm due to the increased entrainment from the large atmospheric turbulence.

The effect of atmospheric turbulence is also clear, both when comparing the general trends of the plot and when intercom-
paring the DTU and FW results for different turbulent intensities. A higher atmospheric turbulence yields a higher production deep inside the farms, while low or even no atmospheric turbulence results in a lower boundary in terms of production.

Finally, the figure includes the resulting power production based on two asymptotic expressions derived by Jensen (1983) and the IWBL model by Frandsen (1992), respectively. The Jensen model is widely used, also by the industry, although it is



less physical as it is not based on a proper momentum analysis. The model yields a velocity ratio given by:

$$\frac{U_\infty}{U_0}\bigg|_{NOJ} = 1 - \frac{2x}{1-x} \quad \text{where} \quad x = \frac{1}{3}\left(\frac{r_0}{r_0 + \alpha x_0}\right)^2 \tag{1}$$

Here, $r_0$ is the turbine radius, $x_0$ is turbine spacing and hence, the original Jensen model only has a single input parameter, $\alpha$, which governs the wake decay and expansion. The recommended values of $\alpha \approx 0.04$ for offshore wind farms, see *e.g.* Barthelmie and Jensen and $\alpha \approx 0.075$ for onshore, see *e.g.* Pena Diaz et al. (2016), are also plotted for reference. The model has previously been compared to CFD simulations in Andersen et al. (2014). The power is computed from the cube of the converged velocity ratio.

The model developed by Frandsen is on the other hand more physical, and involves more parameters, which are interlinked. Here, the expression given in Frandsen and Madsen (2003) is used, which gives the converged velocity at hub height

$$U_{h,Frandsen} = \frac{G}{1 + ln\left(\frac{G}{f' \cdot h}\right)\frac{\sqrt{c_t + (\kappa/ln(h./z_0))^2}}{\kappa}} \tag{2}$$

The geostrophic wind ($G$) and the roughness length ($z_0$) have an impact on the velocity at a given height. Hence, the geostrophic
wind has been calibrated to give a mean wind speed of 8 m/s at a hub height of 90 m for two realistic roughness lengths corresponding to turbulence intensities of 3% and 15%. A latitude of 55° is assumed and a modified parameter of $A_* = 4$ is used for compute $f' = 1.2 \cdot 10^{-4} \cdot exp(A_*)$. The geostrophic wind and roughness lengths are summarized in Table 5. The converged velocity is then found using $C_T = 0.8$ for various distances. The mean power production ratio is then computed using the cube of the converged velocity ratio and assuming constant $C_T$.

**Table 5.** Calibrated geostrophic wind speeds ($G$) and corresponding roughness length ($z_0$), which yield a wind speed of 8 m/s at hub height of 90 m.

| $z_0$ [m] | $G$ [m/s] |
|---|---|
| 0.001 m | 9.99 m/s |
| 1.0 m | 13.59 m/s |

Both models capture the general trends very well, although the Jensen model underestimates the actual power production for the recommended values. The IWBL model by Frandsen performs very well and captures both the high and low turbulent intensity levels as well as the gradual change for the lower turbine spacings where the data by Stevens et al. is located. The simpler models give a good first estimate of the converged mean power production, but the simpler models do not capture the inherent variability of the power production, as the models are merely steady state. The continued importance of developing
and testing such analytical models to provide accurate estimates of both mean velocity and variability was discussed in detail by Meneveau (2019).



**Figure 7.** Mean power production of all turbines from the 6th to the end of the row for all simulations as function of representative turbine spacing. The mean power productions have been normalized by the mean power production of first wind turbine. Errorbars show standard deviation of all the 10 min periods. Simulations with turbulence intensity of 0%, 3%, 10%, and 15% are shown in green, blue, cyan, and red, respectively. Two simulations with $U_0 = 15m/s$ are shown in gray, which have turbulence intensities of 0% and 15%. DTU results are plotted with circles, FW with triangles and UU with squares. Data from Stevens et al. (2015a), which used a constant $C_T = 0.75$ is included for comparison. The underlying broken contours indicate the asymptotic expression (eq. 12) from Jensen (1983) for two different $\alpha$-parameters while the full lines are contours from Frandsen (2007) for two different $z_0$-values.

#### 4.2.2 Response Surface

The total amount of aggregated data in Figure 7 comprise $12,016$ different, albeit overlapping, 10 min realizations, which includes the variability, both within a given 10 min realization and between different 10 min realizations as shown previously.

The power per ground area, or power density, compared to the standard deviation of power normalized by the mean power for different relative spacings is shown in Figure 8, where each black dot is a 10 min realization. The data show significant spread in both power per area and standard deviation of the power although all simulation results generally cluster together. The bin averaged data is shown in red with the standard error plotted as error bars. The standard error is here defined as:

$$\epsilon_{std} = \frac{\sigma\left(\frac{\overline{P}_{10min}}{S_X \times S_Y}\right)}{\sqrt{N}} \tag{3}$$



*i.e.* the standard deviation of the power density within a given bin normalized by the square root of the number of observations. The binned values are generally very consistent except at low standard deviations, in particular Figure 8a) and Figure 8e), where the binned values jump. The standard error is usually large due the a limited number of points, but occasionally it is also small as the limited number of samples are located in small clusters. Furthermore, it appears that the power production per ground area is not very influenced by the standard deviation normalized by mean power.

A multiple linear regression is applied to the full set of bin averaged data with a freestream velocity of $8m/s$ from Figure 8, *i.e.* aggregating all data with comparable $C_T$. The regression is fitted in a least squares sense using the Matlab function "regress"[1]. The regression fits the bin averaged power production per area to the normalized standard deviation of the power production and the relative turbine spacing. The fit is performed to second order, *i.e.* for $\underline{a} = \sqrt{S_X \times S_Y}$ and $\underline{b} = \frac{\overline{P}_{10\min}}{S_X \times S_Y}$ the fit is based on a linear combination of the following matrix and combinations of $a$ and $b$:

$\widetilde{A} =$                                                                                                          (4)

The fit gives coefficients for each combination, and the combined set yields a response surface of the fit.

Figure 9 shows a contour plot of the response surface. The power density found here for a freestream velocity of $8m/s$ is in the range of $0.5 - 2.0W/m^2$, which is comparable to the general range of $1 - 11W/m^2$ reported by Denholm et al.. The power density clearly decreases when the relative turbine spacing increases, as expected, because although the power production

increases for larger spacing, the area increases faster and hence dominates the ratio. However, it is also clear how the power density varies with the standard deviation of the power production, *i.e.* how much power are the turbines able to exploit and extract from the turbulent fluctuations. For large spacing, the power density is not influenced significantly by the standard deviation of the power production, *cf.* Figure 8g). For smaller spacing, there is an increased power density for small standard deviations in the power production, albeit related to the aforementioned small clusters of increased power density for small

spacing, particularly seen in Figure 8a).

The figure also includes circles indicating the binned data used for the fit. The circles are colored according to the difference between the fit and the binned data. The difference is generally less than $\pm 0.5W/m^2$ and alternating between a positive and negative difference for different relative spacings. The fit is particularly good for larger spacings, but it struggles for smaller spacings with large outliers.

As shown previously, the simple engineering models by Frandsen (2007) is capable of capturing the average trends, similar to the response surface. However, the inherent variability of LES is important for farm performance and for improving risk assessment during the design phase. Hence, a similar response surface can be fitted to the standard deviation and the standard error of the bin averaged values in Figure 8. The corresponding response surfaces are shown in Figures 10 and 11, which can be interpreted as the variability and the uncertainty associated with the response surface of mean power density.

The variability around the mean is up to $0.4W/m^2$ for small relative spacings, which is comparable to the difference in the fit and bin averaged data as shown before. The variability is higher for shorter spacings, where the amount of outliers affect the

---

[1]https://se.mathworks.com/help/stats/regress.html



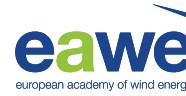
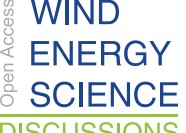

**Figure 8.** Power per ground area plotted against standard deviation of power normalized by mean power for a) $\sqrt{(S_X \times S_Y)} = 10.95R$, b) $\sqrt{(S_X \times S_Y)} = 12.00R$, c) $\sqrt{(S_X \times S_Y)} = 12.65R$, d) $\sqrt{(S_X \times S_Y)} = 14.00R$, e) $\sqrt{(S_X \times S_Y)} = 15.49R$, f) $\sqrt{(S_X \times S_Y)} = 16.73R$, and g) $\sqrt{(S_X \times S_Y)} = 20.00R$. Red dots show bin averaged values including the standard error plotted as error bars.




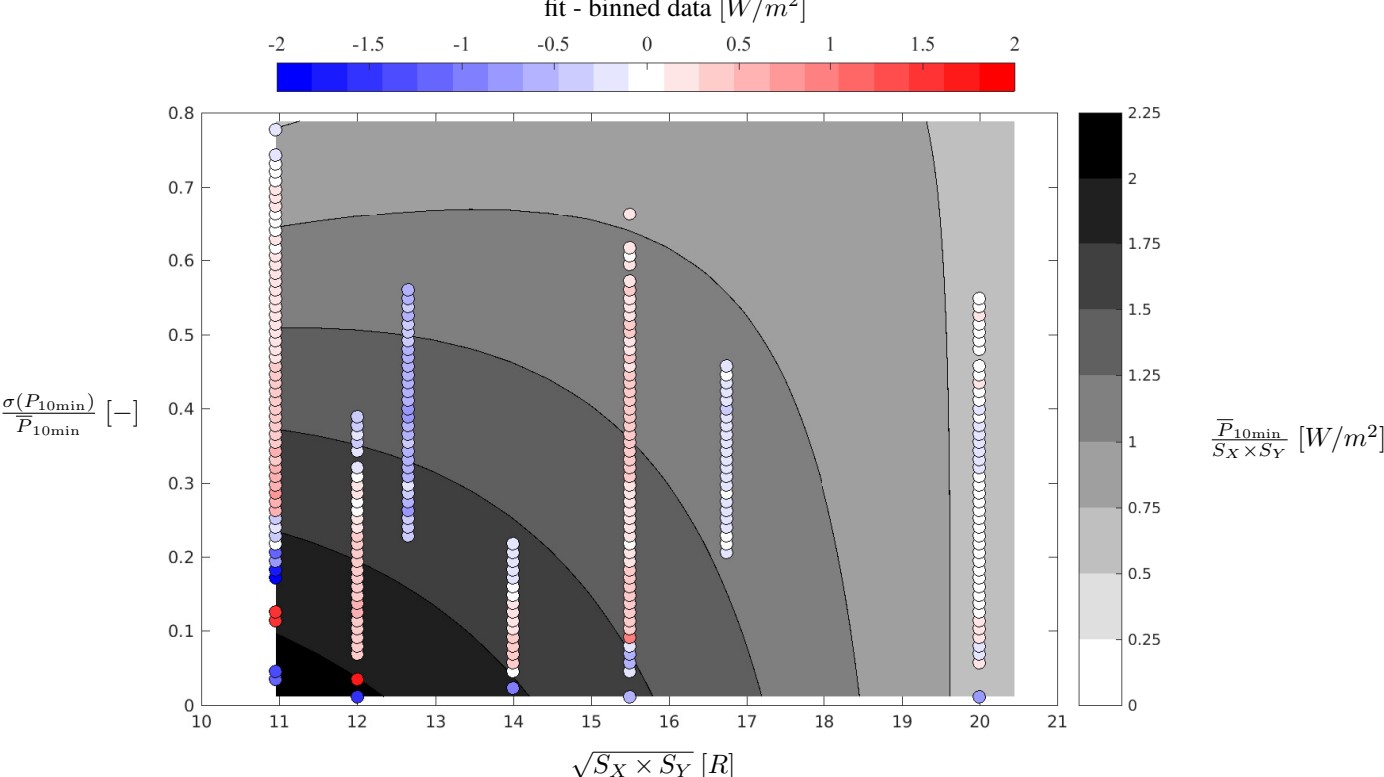

**Figure 9.** Contours based on multiple linear regression fit of bin averaged power production per area to the standard deviation of power production normalized by mean power production and relative turbine spacing. Points mark the binned data, where blue and red shades indicate whether the fit underestimates or overestimates compared to the binned averaged values, respectively.

fit. The outliers can be related to the significant non-linearities in the near wake before the wake breaks down into small scale turbulence, see Sørensen et al. (2015).

The increased variability for smaller spacing also comes with an increased uncertainty as shown in Figure 11. The standard
error decreases for increasing spacing, where the fit is very good, while the discrepancy is larger for the very short distance. The fit tends to overestimate the standard error for the shortest spacings.

The response surfaces are only fitted to second order, because the aim here is merely to provide general insights into the global trends and hence to avoid overfitting. It should be strongly emphasized that this is a rather crude approach. However, the response surfaces yields a first attempt at constructing global response surfaces of the power density including the inherent
variability based on significant amounts of LES data for a wide range of wind farm layouts operating at $8m/s$, which show physical trends.

The response surfaces can continuously be improved by adding more data. Figure 12 shows an occurrence plot of the standard deviation of power production within each 10 min period normalized by the corresponding power production versus the relative turbine spacing. The occurrence plot is based on hexagonal binning of all the data to show the frequency and

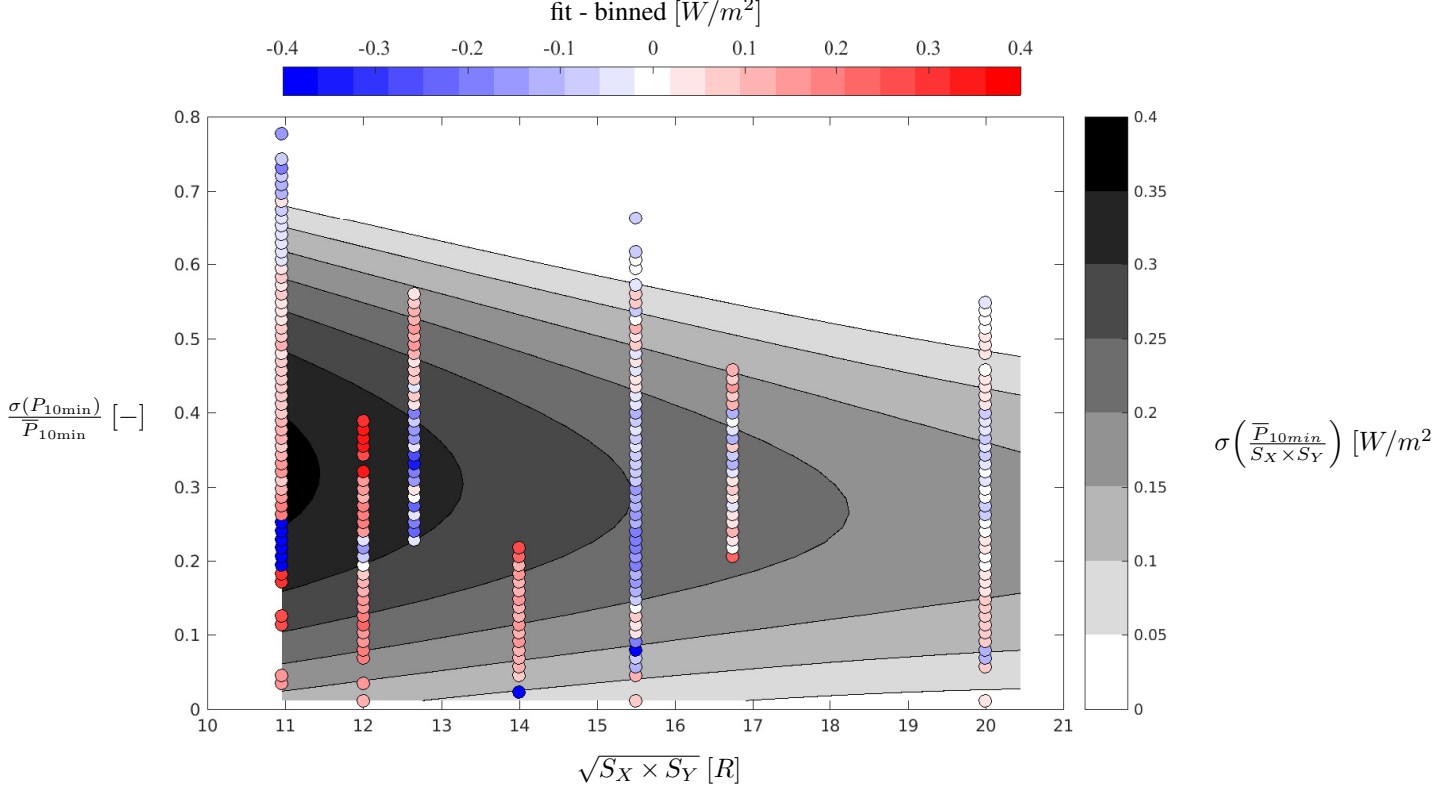

**Figure 10.** Contours based on multiple linear regression fit of standard deviation of the bin averaged power production per area to the standard deviation of power production normalized by mean power production and relative turbine spacing. Points mark the binned data, where blue and red shades indicate whether the fit underestimates and overestimates compared to the binned averaged values, respectively.

location of all the 10 min samples. Clearly, the majority of 10 min realizations are in the range of $\frac{\sigma(P_{10\text{min}})}{\overline{P}_{10\text{min}}} = [0.1 - 0.5]$ for $\sqrt{S_X \times S_Y} = 15.49R$ and $\sqrt{S_X \times S_Y} = 20R$. The overlying dots indicate all $12,016$ realizations for the three contributions, where blue is DTU, green is UU, and red is FW. The trend is clear that the DTU simulations result in smaller standard deviations compared to the FW results. This directly comes from several of the DTU simulations being performed with no atmospheric turbulence, which essentially gives the lower boundary. The results by UU are located in between. This can be used to guide

which scenarios should be computed next essentially to fill the gaps. As such, additional LES computations should be focussed around $\sqrt{S_X \times S_Y} \approx 18R$, where there is no data, and for $\sqrt{S_X \times S_Y} < 14R$, where the uncertainty is large.

The response surfaces could also be made dependent on more parameters by adding more LES data. Currently, the turbine spacings in the lateral and streamwise direction have been collapsed to a single dimension, but the dependency could be unfolded. Similarly, the dependency on a number of additional parameters could be investigated, for instance:

– Free wind speed

      – Turbulence level

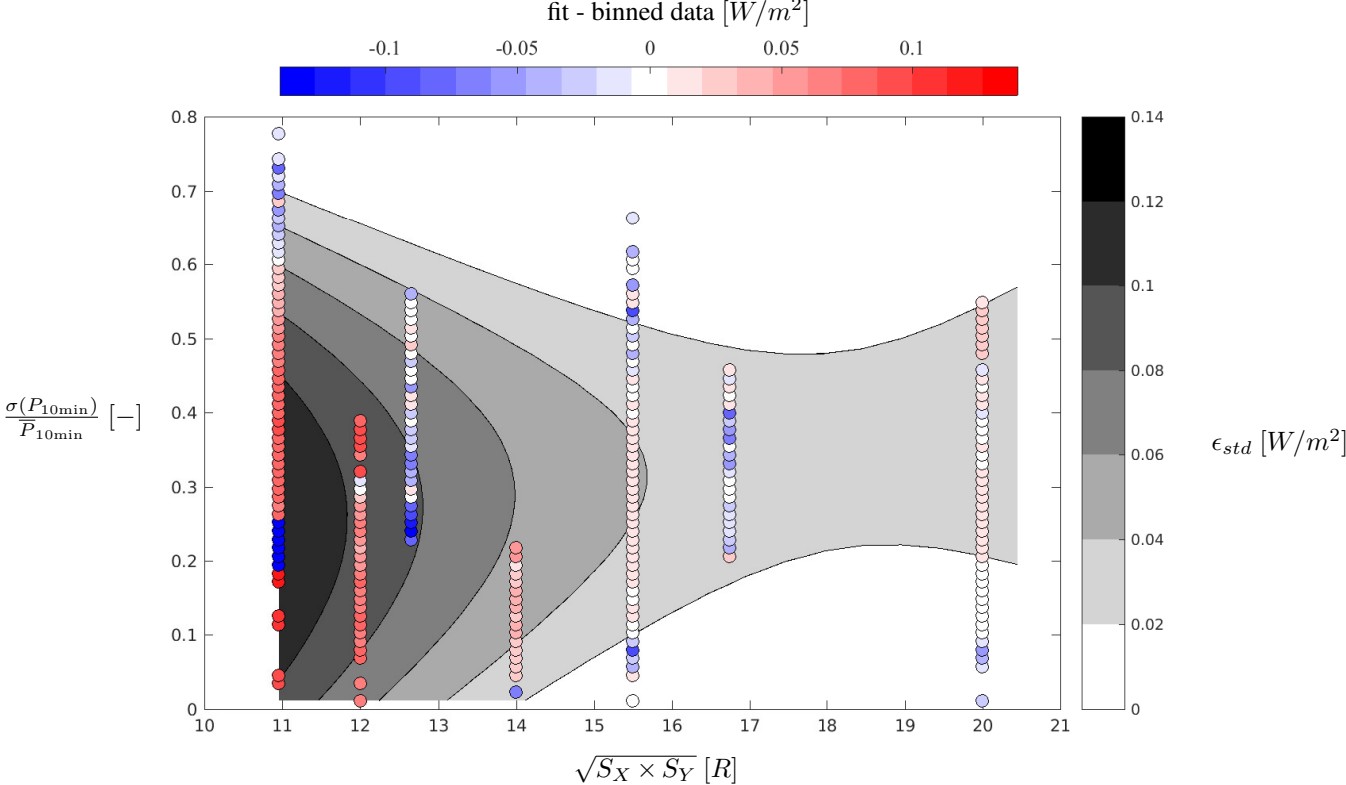

**Figure 11.** Contours based on multiple linear regression fit of the standard error of the bin averaged power production per area to the standard deviation of power production normalized by mean power production and relative turbine spacing. Points mark the binned data, where blue and red shades indicate whether the fit underestimates and overestimates compared to the binned averaged values, respectively.

– Atmospheric stability

– Shear

– Turbine size

However, this would obviously require substantial amounts of computing ressources.

One way to circumvent the large computational costs would be to utilize SCADA data in combination with the LES. Similar response surfaces could be constructed based on SCADA data from operating wind farms, which would enable a more global verification of LES and the actuator disc/line methods on wind farm scale. Such a verification would be valuable as direct comparison of time series of specific events between LES and actual wind farms is extremely difficult, if not impossible, to
achieve given the complexity and amount of information required on the atmospheric conditions to enable such a comparison.

A successful verification would facilitate the direct integration of LES data and SCADA data to construct more certain response surfaces covering a larger range of scenarios and parameters. It could act as a lodestar and inform researchers in



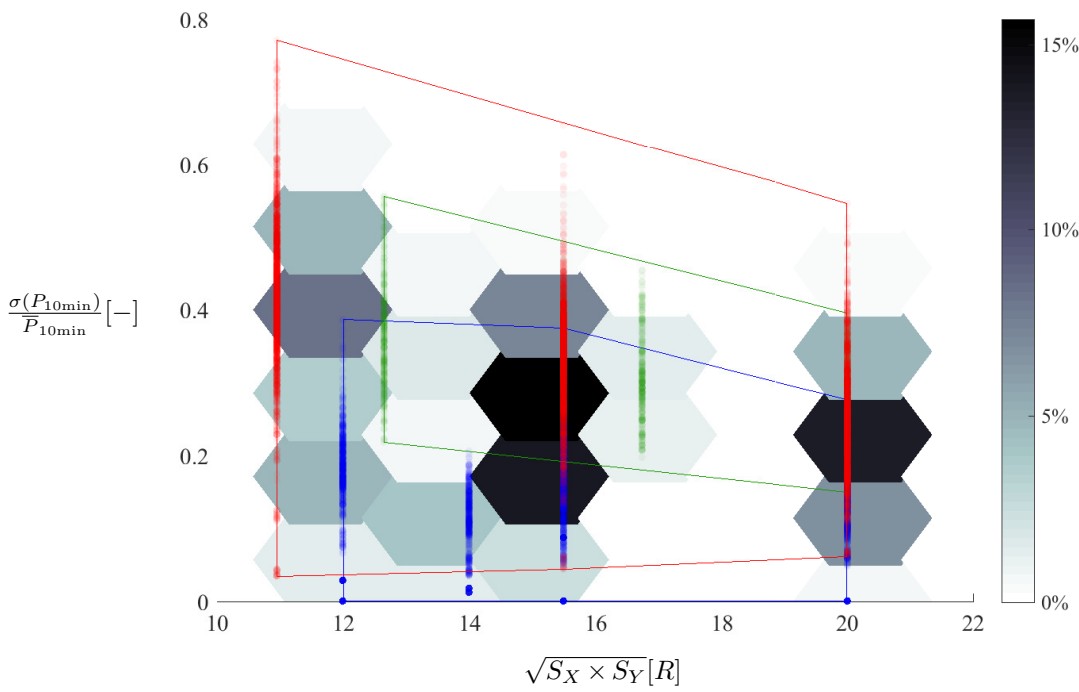

**Figure 12.** Occurrence plot based on hexagonal binning of power production per area against standard deviation of power production based on a total of $12,016$ realizations. Blue dots indicate 10 min realizations from DTU, while green and red are UU and FW results, respectively.

which regions of turbine spacing and turbulence intensity to perform the expensive LES in order to fill the gaps and explain physical trends not captured by the simpler models.

Finally, the response surface could be extended to include *e.g.* fatigue loads for turbines operating in wind farms. Such a surrogate model for fatigue loads on a single wind turbine was developed by Dimitrov et al. (2018), who compared the accuracy and performance of six different methods.

## 5   Conclusions

This work aimed at providing a general overview of the global trends of power performance for large wind farms, with a focus
on variability. This was done through the analysis of Large Eddy Simulations (LES) performed on large wind farms from the three institutions that co-authored this work. LES results of large wind farms obtained from other researchers as well as simulations performed using simpler engineering models were also included to provide a more complete envelope for the results.

As LES require large amounts of computational resources, emphasis was made on extracting as much information possible
from the existing set of simulations performed using different setups and incoming flow conditions. As such, emphasis is not



put on comparing the simulations to each other, but rather on using as many results as possible to cover a wide range of possible scenarios that can provide a global picture of the power characteristics within large wind farms.

Parametric studies were first performed to inform about the effect from atmospheric conditions as well as turbine spacing on the production and its variability. An increase in atmospheric turbulence intensity, by increasing energy entrainment, was shown to raise the mean level of power production. It was also associated to wider distributions of the production values. A larger spacing between the turbines was also associated to greater levels of production, as expected.

The analysis was extended further by aggregating the large amount of LES performed under various conditions. This was done in terms of 10-minute statistics for each turbine operating in deep farm conditions. LES works from other researchers as well as simulations performed with simpler engineering models were also included in a first step when looking at the power produced deep inside the farm as a function of a representative spacing. All results were shown to fall within a clear limit showing how much power can be extracted from a wind farm operating below rated wind speed, as a function of representative turbine spacing. Whereas higher turbulence levels lead to larger production levels deep inside the farms, while cases without incoming turbulence were shown to provide a lower power production. While LES provide more information in terms of variability, simple engineering models were shown to produce a reasonable envelope for the results obtained using the high fidelity methods.

As a second step, response surfaces encompassing the total amount of aggregated LES data, *i.e.* 12,016 different albeit overlapping 10-minute realizations, were created. They revealed information regarding various aspects of the power production within large wind farms, among which the amount of power the turbines are able to extract from the turbulent fluctuations, as well as the variability and uncertainty associated with the mean power densities.

The work presented in this paper serves to provide valuable information regarding power and its variability deep inside large wind farms. Nonetheless, the response surfaces presented here would gain in being complemented with more LES results to provide an even more complete picture. This could be done by considering further turbine spacings to fill existing gaps. The dependency of response surfaces to more parameters could also be investigated, including individually-considered spanwise and streamwise spacings, the freestream velocity as well as the atmospheric stability. As LES are known to be very computationally demanding, SCADA data could also be used to provide more complete response surfaces. Future work could also go one step further by investigating the behavior of turbine loads in similar terms as what was performed here regarding power production.

*Data availability.* The data can be made available on request.



*Author contributions.* S.A., S.-P. B., and B.W. performed the LES. S.A. performed the preliminary analysis and first draft, while all authors
contributed to the further analysis and reporting of the research presented in this manuscript.

*Competing interests.* The authors declare that they have no conflict of interest.

*Acknowledgements.* The work has been carried out with the support of the Danish Council for Strategic Research (grant 2104-09-067216/DSF)
for the project *Center for Computational Wind Turbine Aerodynamics and Atmospheric Turbulence* (COMWIND: http://www.comwind.org),
in the framework of the project *Parallelrechner-Cluster für CFD und WEA-Modellierung* funded by the German Federal Ministry for Eco-
nomic Affairs and Energy (FKZ: 0325220), and with support of the Nordic Consortium on Optimization and Control of Wind Farms funded
by the Swedish Energy Agency. Computer time was granted by the Swedish National Infrastructure for Computing (SNIC) and by the North
German Supercomputing Alliance (HLRN). The proprietary data for Vestas' NM80 turbine has been used. Finally, the authors wish to thank
Juan Pablo Murcia Leon for many fruitful discussions on data analysis.





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
