# Peer review of "Global Trends of Large Wind Farm Performance based on High Fidelity Simulations"

_Wind Energy Science, 2019_

## Short Comment (SC1) · 5 Feb 2020

FYI, The recent finding that a maximum entropy principle probability distribution for wind speeds over large regions follows a BesselK probability distribution. See Pukite, P., Coyne, D. & Challou, D. Mathematical Geoenergy: Discovery, Depletion, and Renewal, Chapter 11 Wind Energy (Wiley, 2019). doi:10.1002/9781119434351.
* * *

---

## Referee Comment (RC1) · Anonymous Referee #1 · 28 Feb 2020

——————-General comments In the manuscript the authors compare the results from different large eddy simulation codes for the performance of very large wind farms. The analysis focuses on the variability of the turbine power production in aligned and staggered wind farms. This is a relevant topic for the community and analysis of this aspect in large eddy simulations is still limited. As indicated at the end of the introduction this study is a continuation of the study by Andersen et al. (2015), but it includes more data and more analysis. The topic addressed in this study, the power variations in wind farms, is an important area that needs further study, relevant for the scientific community, and the readers of Wind Energy Science. Before I can recommend publication of the manuscript, I would like the authors to consider the points indicated below.

——————-Specific comments * In some places the introduction feels a bit like a sum-

[Figure]

mary list of several previous studies as each paragraph summarized the work of one paper. It would be nice when the introduction can be somewhat more coherent.

* There are very few recent papers (last 3 / 4 years) mentioned in the introduction. Please check whether some recent works need to be included in the discussion.

* Figure 2: The Forwind data are for a different turbulence intensity than the other two data sets. As discussed in the manuscript this significantly affects the results. It would be very nice when it would be possible to add one Forwind simulation for the same turbulence intensity as the other cases to allow for a more one to one comparison.

* Section 4.1.1 to 4.1.3 seem to be written rather independently. It would be nice to indicate the connection between these different cases.

* Figure 4 to 6 please give the relevant information necessary to read the figure in the caption or a legend. Now one has to go back to table 1 to find the necessary information to understand the figure. So please mention which mean flow properties are different for each presented data set in these figures.

* Figure 4: For the blue data $P/P\_0=0.32/0.33$ seems a somewhat special value, i.e. there seems to a strong drop in the occurrence of productions that are higher/lower than this particular value. Is there a reason for this?

* In figure 7 there is one gray data point at $\sqrt(s\_ys\_x) \approx 16$, which is much higher than all the other data points. Can the authors discuss this particular cases in more detail.

* Figure 8: Would it be possible to indicate the results for the high and low turbulence intensity cases in different colors, so the effect can be observed and discussed? The figures also have a lot of white space, which can be reduced such that the actual data can be seen better.

* Line 326-329: The authors mention a difference of +-0.5W/m^2. The values in the corresponding plot (figure 9) seem to vary between 0 and 2.25W/m^2. Can the authors

discuss more how this uncertainty should be interpreted?

* Figure 9-11: I am wondering whether the authors can comment in more detail on the uncertainty or potential bias that is introduced by the spread of the available data points over the considered parameter space, which is indicated in figure 12. Is this taken into account in the fitting procedure?

* In figure 2 we have seen that the turbulence intensity influences the performance of the wind farm significantly. In figure 9 to 11 the data for different turbulence intensities are combined. To what degree does this affect the presented results?

* Line 220: Do the authors have an idea on the reason for this increased variability.

————Technical corrections

* line 42: Same reference is mentioned twice. * Equation 4 seems missing. * Table 4: what is meant by "Data is only given for one row of 50 turbines"? * line 178: add as "space" after turbine spacings * line 365: Corrected typo in "ressources" * References: Several references need to be updated – line 467: please update: It is a 2015 paper that is listed as "accepted for publication" – line 481: Spaces are missing in this reference

---

## Referee Comment (RC2) · Anonymous Referee #2 · 15 Mar 2020

General notes

This article brings a comparative analysis of wind farm performances (in terms of mean mechanical power and its variability) based on LES numerical simulations. In the continuity of the work of Andersen et al. (2015), these simulations are performed using 2 codes (EllipSys3D and PALM), 2 rotor modeling methods (Actuator Disk and Actuator Line) and two rotors (NREL 5MW and NM80). Different operating conditions (turbines spacing, mean wind speed, turbulence and shear...) were tested, leading to a total of 18 wind farms cases. The first part of the article results aims to highlight some trends in the influence of operating conditions while the second part aims to show a generelized analysis by aggregating all the results.

This paper brings interesting results which are of importance for the wind energy com-

munity. The objectives of the study are well-posed and the methodology well described. High fidelity LES of wind farms is a state-of-the-art methodology and the obtained results constitute a step forward in the wind farm performances understanding. This topic, still needing further studies, is relevant for Wind Energy Science readers. Nevertheless, some points need to be adressed by the authors before publication.

Specific comments

- 1. Introduction: The introduction makes the impression of being a list of summarized publications. Even if it is well written, some links between articles would be welcome.

- 2.2.1 Ellipsys3D: The aeroelastic coupling may deserve a one-line description to undersand what is involve in the computations (even if is described in the paper of Sorensen et al (2015)).

- 3. Simulation set-up: are the numerical grids cartesian structured?

- 3.1.3 Summary of Numerical Methods: the number of differences between DTU, FW and UU methodogies may constitute a strong difficulty when comparing to each other, specifically between DTU and UU. Additionnaly to the differences given in Tab 1, why are the turbulence and rotor positions different (6R and 10R vs 13R and 30R) for DTU and UU, as well as the total simulation time (60 min vs 30 min)? Even if the authors try to limit their consequences, can the authors can comment on this topic? An identitcal set-up with both codes would help to clarify the code influence for example.

- 3.1.3 As the article deals with high fidelity LES and as it is clearly indicated that such computations are expensive, informations on the computational cost (time step, CPU hours per case, mean reduced computational time...) would be relevant.

- 4.1 Variability of LES: the 40% difference in mechanical power production observed in FW results are assumed to be due to lower turbulence and differences in shear and Coriolis effect treatment. Does the code difference can lead to such gap also?

- 4.1 Variability in LES: What are le LES filering effects involved? The spatial filtering

from the LES approach or the one due to statistical binning?

- 4.1 Even though both plot types is consistent, why the plot type goes from box plot to violin plot by changing the effect influence?

- 4.2.2 Surface Response: all the results presented here are very interesting as well as the type of illustration because gathering so many results is very challenging. I am more concerned on the analysis and moderate it in light of what I indicated for the 3.1.3 point. Can the authors can discuss that?

Technical comments

- Line 42: Stevens et al. (2015) is cited twice in the same sentence.

- Line 67: turbulence -> turbulent

- Line 77: as is -> as it is

- Figure 1: axes unit are missing

- Table 1: the columns need to be explined (U0, ambient TI, shear, turbine resolution)

- Line 202: turbine for are -> turbine are

- Figure 3: the box plots are unclear compared to Figure 2. The boxes are almost not visible.

- Table 5: units are given in the first row and shoudn't be given with the values

- Equation 4 is missing and why a and b are underlined in the relations just before?

- Figure 12: why an hexagonal binning? The white color indicates both a 0% occurence and a lack of data. This should be more distinct.

- The rated power P0 should be namely written in the rotor description

- space before parenthesis are often missing in the text

---

## Author Comment (AC1) · 9 Jun 2020

**Response to reviewers 1 and 2**

We would like to thank the reviewers and public comments for their contributions to improve the article. The comments are relevant and constructive and have contributed to increased quality and readability of the new manuscript. Here we have responded to general and specific comments and indicated modifications. A separate manuscript where changes are highlighted is also attached.

Yours Sincerely,

Jens Nørkær Sørensen, Stefan Ivanell, Björn Witha, Simon-Philippe Breton, and Søren Juhl Andersen

**Reviewer 1**

**General Comments**

*In the manuscript the authors compare the results from different large eddy simulation codes for the performance of very large wind farms. The analysis focuses on the variability of the turbine power production in aligned and staggered wind farms. This is a relevant topic for the community and analysis of this aspect in large eddy simulations is still limited. As indicated at the end of the introduction this study is a continuation of the study by Andersen et al. (2015), but it includes more data and more analysis. The topic addressed in this study, the power variations in wind farms, is an important area that needs further study, relevant for the scientific community, and the readers of Wind Energy Science. Before I can recommend publication of the manuscript, I would like the authors to consider the points indicated below.*

**Specific Comments**

*In some places the introduction feels a bit like a summary list of several previous studies as each paragraph summarized the work of one paper. It would be nice when the introduction can be somewhat more coherent.*

We have rewritten parts of the introduction, and included additional references, as requested below.

*There are very few recent papers (last 3 / 4 years) mentioned in the introduction. Please check whether some recent works need to be included in the discussion.*

Thanks for the comment, we agree. We have included the recent review paper by Porté-Agel et al., 2020. In the review article there is also surprisingly

only a single article published after 2017 with particular relevance to these simulations and the present article, see sections 3.2-3.3. The authors suspect the low number stems from a significant focus on wind farm control across several of the dominant institutes in recent years, and the present article is not focussed on wind farm control. We have however added a couple of additional references on the interaction between farm and atmosphere and vertical staggering as well as the paper by Turner V and Wosnik, 2020.

*Figure 2: The Forwind data are for a different turbulence intensity than the other two data sets. As discussed in the manuscript this significantly affects the results. It would be very nice when it would be possible to add one Forwind simulation for the same turbulence intensity as the other cases to allow for a more one to one comparison.*

Thank you for the suggestion. We agree that it would add value, but this will unfortunately not be possible because our contributor from ForWind has changed jobs in the meantime as also indicated in the contact list.

*Section 4.1.1 to 4.1.3 seem to be written rather independently. It would be nice to indicate the connection between these different cases.*

Thanks for pointing this out. A sentence explaining the connection between the sections has been added and we have improved the segues betweeen the sections.

*Figure 4 to 6 please give the relevant information necessary to read the figure in the caption or a legend. Now one has to go back to table 1 to find the necessary information to understand the figure. So please mention which mean flow properties are different for each presented data set in these figures.*

We have improved the captions.

*Figure 4: For the blue data $P/P_0 = 0.32/0.33$ seems a somewhat special value, i.e. there seems to a strong drop in the occurrence of productions that are higher/lower than this particular value. Is there a reason for this?*

This is due to the controller as written briefly in the submitted article. We have expanded the explanation in the article and included Figure 1 here, which show the normalized power vs rotational speed. This shows that $P/P_0 = 0.32/0.33$ occurs as the rotational speed reaches its maximum.

[Figure]

Figure 1: Normalized power vs rotational speed for DTU3.

*In figure 7 there is one gray data point at $\sqrt{(s_y s_x)} \approx 16$, which is much higher than all the other data points. Can the authors discuss this particular cases in more detail.*

These two simulations($DTU4$ and $DTU6$) are performed with a freestreem velocity of $15m/s$, which is above rated for the first turbine. They form an upper limit, because they are right on the boundary of whether or not the deep wind farm is above or below rated. This is already discussed in 4 lines, but we have added another sentence, which hopefully clarifies further. The article now reads: *"An upper limit is indicated by DTU4 and DTU6 (in grey), which have a freestream velocity above rated (15m/s), but with different turbulence intensities. The power productions deep inside the farm result in below rated conditions for DTU6 due to no freestream turbulence, while the turbines in DTU4 also experience above rated velocities deep inside the farm due to the increased entrainment from the large atmospheric turbulence. Hence, it shows the transition from below rated to above rated conditions."*

*Figure 8: Would it be possible to indicate the results for the high and low turbulence intensity cases in different colors, so the effect can be observed and discussed? The figures also have a lot of white space, which can be reduced such that the actual data can be seen better*

Thanks for a very good suggestion. We have updated the figure accordingly, and colored the dots in green for low turbulence($0 - 3\%$) and red for high

[Figure]

Figure 2: CDF of binned data indicating underestimation and overestimation of the fit.

turbulence$(10 - 15\%)$. It highlights an interesting aspect that the standard deviation of the power seems largest for the small atmospheric turbulence. We have reduced the white space slightly, but we prefer to keep the same scale on all figures to facilitate easier comparison.

*Line 326-329: The authors mention a difference of +-0.5W/m^2. The values in the corresponding plot (figure 9) seem to vary between 0 and 2.25W/m^2. Can the authors discuss more how this uncertainty should be interpreted?*

We have clarified our statement to quantify that "generally" corresponds to 87%. Please see Figure 2 here in the response, which shows the CDF of under- and overestimation.

*Figure 9-11: I am wondering whether the authors can comment in more detail on the uncertainty or potential bias that is introduced by the spread of the available data points over the considered parameter space, which is indicated in figure 12. Is this taken into account in the fitting procedure?*

Thanks for the very good suggestion. We have now performed a k-fold with k = 10 to estimate the uncertainty in fitting the response surfaces, e.g. lines 355-358. As shown the MSE is consistent throughout with only minor variations between the 10 k-folds, but it also indicates as previously stated that it is a rather crude fit, which could be improved in future work.

*In figure 2 we have seen that the turbulence intensity influences the performance of the wind farm significantly. In figure 9 to 11 the data for different turbulence intensities are combined. To what degree does this affect the presented results?*

Thanks for the question. The updated Figure 8, where we have now indicated low and high turbulence, show that the influence is minor on the binned data. The largest effect is seen in Figure 8a), but with significant more data this split could be made.

*Line 220: Do the authors have an idea on the reason for this increased variability.*

This is a complex question. This setup with 50 turbines is rather unique and comparisons are difficult. However, the authors do believe that there might be additional deep farm effects combined possibly with a fully developed boundary layer and potential gravity waves. However, further investigations are needed to make any conclusions on this.

**Technical corrections**

*line 42: Same reference is mentioned twice.*

Fixed

*Equation 4 seems missing.*

Thanks, we have now included the equation.

*Table 4: what is meant by "Data is only given for one row of 50 turbines"?*

The ForWind simulations were performed with 2(two) rows of 50 turbines, however, only data from one of the rows has been used in the analysis. We have rephrased slightly for hopefully improved clarity.

*line 178: add as "space" after turbine spacings*

Fixed

*line 365: Corrected typo in "ressources"*

Fixed.

*References: Several references need to be updated*

Thanks, we realize there were some issues with it. It has now been cleaned up.

*line 467: please update: It is a 2015 paper that is listed as "accepted for publication"*

Fixed

*line 481: Spaces are missing in this reference*

Fixed.

**Reviewer 2**

**General notes**

*This article brings a comparative analysis of wind farm performances (in terms of mean mechanical power and its variability) based on LES numerical simulations. In the continuity of the work of Andersen et al. (2015), these simulations are performed using 2 codes (EllipSys3D and PALM), 2 rotor modeling methods (Actuator Disk and Actuator Line) and two rotors (NREL 5MW and NM80). Different operating conditions (turbines spacing, mean wind speed, turbulence and shear...) were tested, leading to a total of 18 wind farms cases. The first part of the article results aims to highlight some trends in the influence of operating conditions while the second part aims to show a generelized analysis by aggregating all the results. This paper brings interesting results which are of importance for the wind energy community. The objectives of the study are well-posed and the methodology well described. High fidelity LES of wind farms is a state-of-the-art methodology and the obtained results constitute a step forward in the wind farm performances understanding. This topic, still needing further studies, is relevant for Wind Energy Science readers. Nevertheless, some points need to be adressed by the authors before publication.*

**Specific Comments**

*1. Introduction: The introduction makes the impression of being a list of summarized publications. Even if it is well written, some links between articles would be welcome.*

Thanks. This was also addressed by our other reviewer and we have now expanded our introduction.

*2.2.1 Ellipsys3D: The aeroelastic coupling may deserve a one-line description to undersand what is involve in the computations (even if is described in the paper of Sorensen et al (2015)).*

We have expanded the description slightly.

*3. Simulation set-up: are the numerical grids cartesian structured?*

Yes, the grids are cartesian. We have specified this in Section 3 Simulation Setup.

*3.1.3 Summary of Numerical Methods: the number of differences between DTU, FW and UU methodogies may constitute a strong difficulty when comparing to each other, specifically between DTU and UU. Additionnaly to the differences given in Tab 1, why are the turbulence and rotor positions different (6R and 10R vs 13R and 30R) for DTU and UU, as well as the total simulation time (60 min vs 30 min)? Even if the authors try to limit their consequences, can the authors can comment on this topic? An identitcal set-up with both codes would help to clarify the code influence for example.*

The initial benchmark scenarios were defined to perform such a comparison, but it quickly became clear that it is not trivial to perform direct code-to-code comparisons of such large simulations, because certain models could not adhere to the initial definitions. Code-to-code comparisons involves all model dependencies of SGS model, turbine model, numerical schemes etc. So although we agree that code-to-code comparisons are indispensable, we think it should initially be done on simpler scenarios. Hence, the aim of the present article is rather to show the type of global analyses we can perform by combining results from various institutes. We believe we have addressed this with our previous

statement: *"Aggregating the statistics from different simulations and numerical setups essentially assumes that all simulations are physically correct and correspond to different farms/turbines operating under different atmospheric conditions."* We think that the analyses presented in this article show that the results are coherent, when scaled properly, despite e.g. different turbine models..

*3.1.3 As the article deals with high fidelity LES and as it is clearly indicated that such computations are expensive, informations on the computational cost (time step, CPU hours per case, mean reduced computational time...) would be relevant.*

Most of the performed simulations were executed a few years back and we do not think that the actual numbers would be representative anymore. However, we have added an explanation on general levels of computational costs for the used types of simulations relevant for today's computer resources, see section 3.1.3.

*4.1 Variability of LES: the 40% difference in mechanical power production observed in FW results are assumed to be due to lower turbulence and differences in shear and Coriolis effect treatment. Does the code difference can lead to such gap also?*

The large deviation is expected to mainly depend on the TI but also other sources can affect the result. However, the aim here is not to perform a code-to-code comparison and as stated we assume model results included in the global analyses to be physically correct. Of course this includes some uncertainties but these uncertainties will be decreased with more data being included into the developed methodology here presented.

*4.1 Variability in LES: What are le LES filering effects involved? The spatial filtering from the LES approach or the one due to statistical binning?*

We meant the spatial filtering effect from the turbines themselves. We have clarified this now.

*4.1 Even though both plot types is consistent, why the plot type goes from box plot to violin plot by changing the effect influence?*

We wanted to show how the distributions are capped due to the controller in Figure 4, which would not be seen in a box plot. Please see Figure 1 and response to other reviewer.

*4.2.2 Surface Response: all the results presented here are very interesting as well as the type of illustration because gathering so many results is very challenging. I am more concerned on the analysis and moderate it in light of what I indicated for the 3.1.3 point. Can the authors can discuss that?*

Thanks for your questions. Motivated by your question and comment by other reviewer we have now performed a k-fold cross-validation, which examines the sensitivity of the fitted response surfaces. .

**Technical comments**

*Line 42: Stevens et al. (2015) is cited twice in the same sentence.*

Fixed.

*Line 67: turbulence -¿ turbulent*

Fixed.

*Line 77: as is -¿ as it is*

Fixed.

*Figure 1: axes unit are missing*

Fixed.

*Table 1: the columns need to be explined (U0, ambient TI, shear, turbine resolution)*

We believe the referee meant Table 2-4, so we have added an explanation of U0, TI, shear and resolution. Additionally, we have specified in Table 1 that R is turbine radius and $z_{hub}$ is hub height.

*Line 202: turbine for are -¿ turbine are*

Fixed.

*Figure 3: the box plots are unclear compared to Figure 2. The boxes are almost not visible.*

The width of the boxes has been increased.

*Table 5: units are given in the first row and shoudn't be given with the values*

Fixed.

*Equation 4 is missing and why a and b are underlined in the relations just before?*

Thanks, this was a clear oversight. The equation is now included and updated for consistency.

*Figure 12: why an hexagonal binning? The white color indicates both a 0% occurence and a lack of data. This should be more distinct.*

Thanks for the feedback. We have decided to replace the hexagonal binning by a heatmap, which more clearly shows exactly how many realizations we have and where the lack of data is across the parameter space.

*The rated power P0 should be namely written in the rotor description*

P0 has been defined now in Section 2.4

*space before parenthesis are often missing in the text*

This has been fixed.

**Public comment by Paul Pukite**

Dear Paul,
Thanks for your comment and interesting reference. However, in the present article "global trends" refers to our efforts to extract and quantify overall behaviour of large wind farms simulated using LES, and not "global" in terms of world wide or large regions.
Best regards,

The authors

---

## Author Response (AR2)

**Response to 2nd review**

Dear Reviewer,

Once again, thank you for your comments and corrections, which we've now implemented. The newest corrections are shown in red.

Yours Sincerely,

Jens Nørkær Sørensen, Stefan Ivanell, Björn Witha, Simon-Philippe Breton, and Søren Juhl Andersen

**Reviewer 1**

**Specific Comments**

*Line 45. Important work on blockage is also done by Segalini and Dahlberg, Wind Energy, 23(2), 120-128 (2020).*

Thanks, we have added this relevant reference.

*Line 47. Other reviews related to the present work include Duckworth, Barthelmie, Wind Engineering 32(5), 459–475 (2008) Stevens and Meneveau, Annu. Rev. Fluid Mech. 2017. 49:311–39*

Thanks again, we have added these references as well.

*Line 72 Continuous work by Andersen et al. (2015) ==> I am not sure what exactly is meant by continuous work. The preceding sentence discusses Andersen et al. (2016) so the order is confusing here*

Thanks, we have rephrased for clarity.

*Line 100: Specify the participating groups here*

We have specified the participating research groups.

*Line 168: Double check "prescribed boundary layer"; the abbreviation PBL is generally used for planetary boundary layer, so perhaps consider not using the abbreviation.*

We are aware of this potentially confusing abbreviation. However, the use of

PBL is clearly defined and consistent within the article, and we prefer not to introduce a new name for an existing method, when it is not the focus of the present work.

*Table 1: Define the different participating institutions here*

We have defined the institutes.

*Table 2: How is shear defined here / indicate units.*

We have expanded the explanation of the shear coefficient in Section 3.1.1.

*line 213: The authors explained that they were not in the position to perform all simulations for the same turbulence intensity. That is fine. However, in line 213 the differences between the various cases should be specified for clarity.*

Here, we compare three specific cases based on their similarities, but we have rephrased slightly to highlight again that the different methods yield different vertical profiles.

*Line 256: Is this also the reason the numbers in figure 5 are lower than in figure 4?*

Good observation, but the lower numbers in Figure 5 compared to Figure 4 appears consistent with Figure 2, where the power productions seen in the FW simulation is lower than those in the DTU.

*Section 4.2.1: It should perhaps be mentioned that the authors just consider 2 example models. Over the years various other models have already been developed.*

The additional of the suggested references by the reviewer shows that numerous simple wake models exist, so we have kept the previous formulation here.

*Figure 12: Does the clustering of sample points affect the obtained fit? I also asked this question before, but did not get a full answer on this. When this is not fully known perhaps just clarify this around line 380.*

We have extended our explanation, which hopefully makes it very clear that the response surfaces are directly dependent on the available data and that

adding more data would improve the confidence in the response surfaces. It would also enable for the creation of more advanced surrogates using for instance polynomial chaos expansion, neural networks or other methods.

**Technical corrections**

*Please double check the manuscript for typos and use of single / plural, etc.*

Thanks, we have gone through the manuscript again and made corrections, when found.

*Line 15 wind farm(s).*

Fixed.

*Line 37 (and other places) Porte-Agel needs a accent on the e*

Thanks for spotting this, we had a typo in our bibtex.

*Line 68 "of the 10 turbine row"*

It is unclear to us, what the suggested correction is?

*Line 71 "5th of 6th turbine" ==¿ specific that this concerns rows*

We have specified this.

*Line 230-235: Consider combining the different paragraphs*

Done.

*Line 247 double dot at the end of the sentence*

Fixed

*Line 294 IWBL model is not defined*

Thanks. We have decided to denote the two engineering models as the Jensen and the Frandsen model, respectively. The IWBL (internal wind farm boundary layer) was an internal abbreviation, not used by Frandsen in his 1992 paper.

*Line 323 "turbulence(TI" ==¿ add space*

Fixed

*Line 355 "rror(MSE)" ==¿ add space*

Fixed

*Line 392 "textcolorredresources."*

Fixed